# LD-RoViS: Training-free Robust Video Steganography for Deterministic Latent Diffusion Model

Xiangkun Wang[1,2]    Kejiang Chen[1,2]*    Lincong Li[1,2]    Weiming Zhang[1,2]    Nenghai Yu[1,2]

[1]University of Science and Technology of China, China
[2]Anhui Province Key Laboratory of Digital Security, China
wangxiangkun@mail.ustc.edu.cn    chenkj@ustc.edu.cn

## Abstract

Existing video steganography methods primarily embed secret information by modifying video content in the spatial or compressed domains. However, such methods are prone to distortion drift and are easily detected by steganalysis. Generative steganography, which avoids direct modification of the cover data, offers a promising alternative. Despite recent advances, most generative steganography studies focus on images and are difficult to extend to videos because of compression-induced distortions and the unique architecture of video generation models. To address these challenges, we propose LD-RoViS, a training-free and robust video steganography framework for the deterministic latent diffusion model. By modulating implicit conditional parameters during the diffusion process, LD-RoViS constructs a dedicated steganographic channel. Additionally, we introduce a novel multi-mask mechanism to mitigate errors caused by video compression and post-processing. The experimental results demonstrate that LD-RoViS can embed approximately 12,000 bits of data into a 5-second video with an extraction accuracy exceeding 99%. Our implementation is available at `https://github.com/xiangkun1999/LD-RoViS`.

## 1   Introduction

In the contemporary digital era, the rapid spread and extensive sharing of information have brought increasing attention to information security. Steganography [1], as a key technique for information hiding, aims to embed secret data within normal data to enable covert communication. Among various media, video has become a particularly valuable cover for steganography because of its high complexity and large capacity for hiding data [2]. In contrast, steganalysis [3], as an adversarial technique of steganography, seeks to detect the presence of hidden information.

Early video steganography methods can be broadly categorized into spatial domain methods and transform domain methods. Spatial domain methods [4, 5, 6, 7, 8, 9] embed information by directly modifying pixel values in video frames. Transform domain methods [10, 11, 12, 13, 14, 15, 16], on the other hand, leverage transformations such as DWT and DCT to embed data in the frequency domain, enhancing robustness against video encoding. Given the advantages of compressed video in storage and transmission, researchers have also explored compressed domain steganography [17, 18, 19, 20, 21, 22, 23], embedding messages into elements such as DCT coefficients, motion vectors, and prediction modes during video compression. While effective in certain scenarios, these traditional methods inevitably modify video content or encode content, making them vulnerable to distortion drift and steganalysis attacks [24, 25, 26].

With the rapid development of generative AI, many high-performance models has emerged, capable of generating realistic text, images, and video. Among them, video generation models based on

---

*Corresponding author.

39th Conference on Neural Information Processing Systems (NeurIPS 2025).

diffusion models have become a dominant paradigm, exemplified by advanced systems such as Sora [27], Gen-4 [28], Veo-2, HunyuanVideo [29], and Wan2.1 [30]. Simultaneously, AI-generated videos have become widespread on social media platforms, reshaping the data environment for video steganography. According to statistics from Zebracat AI[2], AI-generated videos account for 40% of video content on major social platforms. Recent studies have proposed generative steganography methods [31, 32, 33, 34, 35, 36, 37, 38, 39], which avoid directly modifying the cover data, and instead embed secret messages implicitly during the data generation process. These methods have shown strong resistance to steganalysis. However, current research in this area has focused almost exclusively on images, leaving generative video steganography largely unexplored.

Given that videos are composed of sequences of images, and that generative image steganography has matured considerably, an intuitive question arises: Can these methods be extended to videos? Some researchers have made early attempts in this direction. For example, [40] extends [31] into latent space and provides a simple implementation of video steganography based on Stable Video Diffusion [41]. Similarly, [42] employs a face-swapping model to embed messages into facial features, achieving robust video steganography. However, these methods are not compatible with most state-of-the-art (SOTA) video generation models, which face three main challenges: First, SOTA video generation models often rely on deterministic sampling to accelerate inference, making methods such as [31, 32, 33, 40]—which depend on random noise sampling—inapplicable. Second, most diffusion samplers are designed for one-directional generation and cannot achieve perfect invertibility, whereas [36, 37, 38, 39] map messages to the initial noise and depend on accurate inversion of the sampling process for extraction, making them unsuitable for non-reversible video diffusion models. Third, compression processing [43] imposed by video-sharing platforms presents additional challenges to the robustness of steganographic embedding.

To address these limitations, we propose LD-RoViS (**L**atent **D**iffusion-based **Ro**bust **Vi**deo **S**teganography), a training-free and robust steganography framework designed for deterministic latent diffusion models. By modulating the conditional parameters at the final time step of the diffusion process, we construct a steganographic channel to embed secret information into latent variables. Furthermore, to ensure robustness against video compression on social platforms, we introduce a multi-mask mechanism to identify and utilize the robust regions in the latent space for message embedding.

Our main contributions can be summarized as follows:

- We introduce LD-RoViS, the first training-free video steganography method for deterministic latent diffusion models.

- We design an implicit parameter modulation strategy to seamlessly integrate message embedding into the generation process.

- We propose a multi-mask mechanism to identify robust regions in latent space, enabling adaptive message embedding and enhancing resistance to compression-induced errors.

- We conduct extensive experiments to evaluate LD-RoViS, demonstrating its superior performance over existing methods in terms of capacity, robustness, and security.

## 2 Related Work

### 2.1 Diffusion Models

The core framework of diffusion models consists of two phases: forward diffusion and reverse denoising. The forward process incrementally corrupts data with Gaussian noise until it becomes pure noise, whereas the reverse process learns to predict and remove noise via neural networks to reconstruct the original data. Diffusion models have emerged as the dominant paradigm in generative models because of their high-quality generation capabilities. Current diffusion models can be categorized on the basis of their sampling strategies, as outlined below.

**DDPM (Denoising Diffusion Probabilistic Model).** The forward process of DDPM [44] follows a Markov chain for noise addition, whereas the reverse process trains a U-Net via variational inference.

---

[2] https://www.zebracat.ai/post/ai-video-creation-statistics

From one time step $t$ to the previous time step $t-1$, the reverse sampling in DDPM can be expressed as:

$$\mathbf{x}_{t-1} = \frac{1}{\sqrt{\alpha_t}}\left(\mathbf{x}_t - \frac{1-\alpha_t}{\sqrt{1-\bar{\alpha}_t}}\,\boldsymbol{\epsilon}_\theta(\mathbf{x}_t, t)\right) + \sigma_t\epsilon, \epsilon \sim \mathcal{N}(0, \mathbf{I}), \tag{1}$$

where $\boldsymbol{\epsilon}_\theta(\mathbf{x}_t, t)$ is the noise predicted by the neural network, $\alpha_t$ and $\bar{\alpha}_t$ are predefined noise scheduler parameters, and $\sigma_t$ is the standard deviation of the sampled Gaussian noise. Despite excellent generation quality, DDPM requires hundreds of sampling iterations, leading to low efficiency. Additionally, its reliance on stochastic sampling of $\epsilon$ makes it a non-deterministic model.

**DDIM (Denoising Diffusion Implicit Model).** To address DDPM's inefficiency, DDIM [45] employs a non-Markovian process and reparameterization to enable sampling with arbitrary step sizes. From a time step $t$ to a time step $s$ (where $s < t$), the DDIM sampling can be expressed as:

$$\mathbf{x}_s = \sqrt{\bar{\alpha}_s}\mathbf{f}_\theta(\mathbf{x}_t, t) + \sqrt{1-\bar{\alpha}_s - \sigma_s^2}\boldsymbol{\epsilon}_\theta(\mathbf{x}_t, t) + \sigma_s\epsilon, \ \mathbf{f}_\theta(\mathbf{x}_t, t) = \frac{\mathbf{x}_t - \sqrt{1-\bar{\alpha}_t}\boldsymbol{\epsilon}_\theta(\mathbf{x}_t, t)}{\sqrt{\bar{\alpha}_t}}, \tag{2}$$

where $\boldsymbol{\epsilon}_\theta(\mathbf{x}_t, t)$ is the noise predicted by the neural network, $\bar{\alpha}_t$ and $\bar{\alpha}_s$ are the predefined noise scheduler parameters. When $\sigma_s \neq 0$, DDIM becomes non-deterministic; otherwise, DDIM is deterministic.

**Flow Matching.** Instead of relying on Markov chains, Flow Matching [46] formulates diffusion as the continuous evolution of vector fields. From a time step $t$ to a time step $t - \Delta t$, the flow-based sampling in Flow Matching can be expressed as:

$$\mathbf{x}_{t-\Delta t} = \mathbf{x}_t - \Delta t \cdot \mathbf{f}_\theta(\mathbf{x}_t, t), \tag{3}$$

where $\mathbf{f}_\theta(\mathbf{x}_t, t)$ is an output of a neural network. Since Flow Matching does not involve the sampling of random variables, it is deterministic.

## 2.2 Diffusion-Based Generative Steganography

Steganographic methods based on diffusion models embed hidden information into the data synthesis process. Previous work mainly focused on image steganography, which supports image-level [34, 35] or message-level [31, 40, 33, 32] hiding. In this paper, we focus on message-level hiding.

**StegaDDPM** [32] performs steganographic embedding at the final time step of the reverse diffusion process. By partitioning the probability density function of the Gaussian distribution into equal-probability intervals, it constructs a reversible mapping from binary messages to Gaussian noise variables $\epsilon$, thereby enabling message embedding within the generated images. However, this method is highly sensitive to lossy image transformations.

**LDStega** [33] extends StegaDDPM from the pixel space to the latent space, making it compatible with mainstream latent diffusion models. By mapping messages to a biased distribution, LDStega reduces the probability of extraction errors. In addition, it enhances robustness by performing a pre-encoding and decoding scheme on the latent variables.

**Pulsar** [31] adopts a different mapping strategy. At the final time step of the diffusion process, two noise samples, $\epsilon_1$ and $\epsilon_2$, are generated via two distinct secret keys. These are then mixed according to a binary message to produce the final noise added to the image. The receiver extracts the hidden message by comparing the distances between the stego image and the two reference images generated via the secret keys.

**PSyDUCK** [40] extends Pulsar from the pixel space to the latent space and investigates the impact of using two distinct keys at different diffusion timesteps. Moreover, it explores the applicability of the method in video generation models.

These methods heavily rely on non-deterministic sampling models. However, mainstream video generation models typically adopt deterministic sampling to enable faster inference and better controllability [30], rendering the above methods inapplicable. Moreover, the compression [43] applied by video-sharing platforms poses additional challenges to the robustness of video steganography methods.

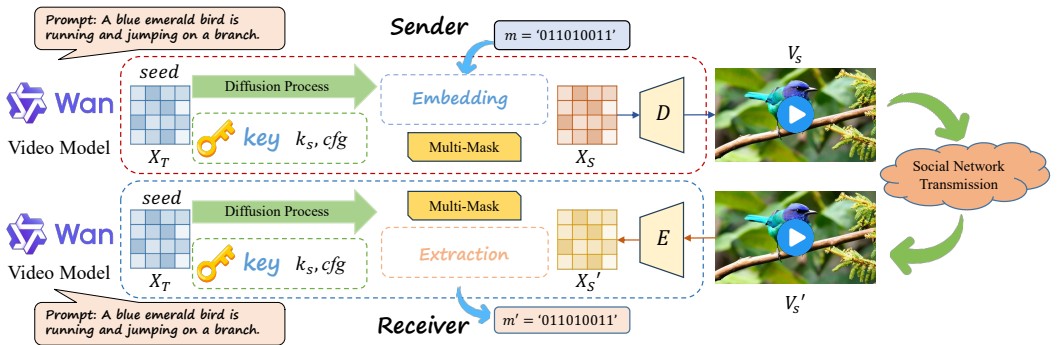

Figure 1: Overview of our proposed LD-RoViS.

## 3 Method

Our robust video steganography method is built upon a deterministic latent diffusion video model, enabling reliable message embedding and extraction even under challenging transmission conditions. The following sections provide a systematic overview of the proposed method.

### 3.1 Overview of LD-RoViS

As shown in Fig. 1, the sender and receiver share the same parameters: $prompt$, $seed$, $k_s$, and cfg, where $prompt$ refers to the text input fed into the video generation model to control the semantic content of the generated video. $seed$ is a random seed that controls the sampling of the initial noise $X_T$. The values $k_s$, and cfg denote different values of the classifier-free guidance (CFG) scale. The CFG scale controls how closely the generated content aligns with the input prompt.

On the sender's side, $prompt$, $seed$, and $k_s$ are used to drive the diffusion process in the latent space up to the final time step. At this final time step, message embedding is performed via cfg and the message $m$, yielding the final latent variable $X_s$. This latent variable is then passed through the decoder $D$ to generate the stego video $V_s$. The stego video is shared with the receiver via social platforms, and the received video is denoted as $V_s'$. The receiver encodes $V_s'$ via the encoder $E$ to obtain the latent variable $X_s'$, and leverages the shared $prompt$, $seed$, and $k_s$ to perform the same generation procedure as the sender. At the final time step, the receiver uses cfg and $X_s'$ to extract the hidden message.

In the following, we use $V \in \mathbb{R}^{C \times F \times H \times W}$ to represent the video, and $X \in \mathbb{R}^{C' \times F' \times H' \times W'}$ to represent the latent variable, where $C, F, H, W$ denote the number of channels, frames, height, and width of the video $V$, while $C', F', H', W'$ represent the corresponding attributes in the latent space.

### 3.2 Message Embedding

The detailed framework of our proposed LD-RoViS is shown in Fig. 2, where "Shared" denotes the operations that need to be performed by both the sender and the receiver.

To embed a message, the sender first executes the "Shared" component. Specifically, the sender provides $prompt$ and $seed$ to the video generative model (denoted as $G$). The model $G$ begins by sampling $X_T$ from a Gaussian distribution using the given $seed$, and then performs $T - 1$ denoising steps to obtain the latent representation $X_1$. We use $Diffuse(\cdot)$ to represent the denoising process, then the above process can be formulated as:

$$X_1 = Diffuse(G, X_T, k_s). \tag{4}$$

#### 3.2.1 Parameter Modulation Strategy

The parameter modulation strategy lies at the core of our steganographic method. During each denoising step, the CFG scale controls the weighting between the predicted conditional and unconditional noise, formulated as:

$$\epsilon_\theta(\mathbf{x}_t, t) = pred_{uncond} + \text{CFG} \cdot (pred_{cond} - pred_{uncond}), \tag{5}$$

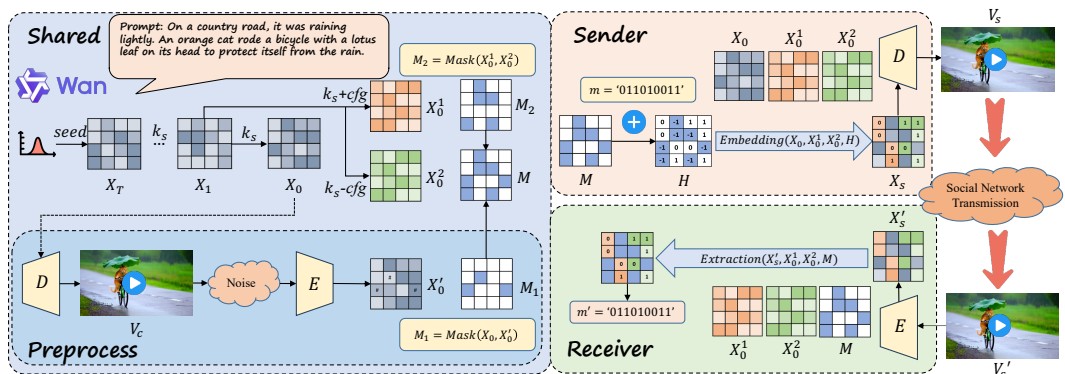

Figure 2: Framework of our proposed LD-RoViS.

where $\epsilon_\theta(\mathbf{x}_t, t)$ denotes the noise predicted by $G$ at each timestep, while $pred_{cond}$ and $pred_{uncond}$ represent the conditional and unconditional predicted noise, respectively.

Varying the CFG scale alters the trajectory of the diffusion process, resulting in different latent variables. This modulation strategy is equally applicable to deterministic diffusion models. On the basis of this insight, we can generate three different outputs—$X_0$, $X_0^1$, and $X_0^2$—from the same latent variable $X_1$ in the final denoising step by using different values of CFG $k_s$, $k_s + \text{cfg}$, and $k_s - \text{cfg}$, respectively. This can be expressed as:

$$X_i = Diffuse(G, X_1, k_i), \quad \text{where } X_i \in \{X_0, X_0^1, X_0^2\},\ k_i \in \{k_s, k_s + \text{cfg}, k_s - \text{cfg}\}. \quad (6)$$

These latents exhibit subtle semantic differences while maintaining visual consistency with $prompt$, creating a discriminative latent pair for message encoding.

### 3.2.2 Multi-mask Mechanism

Before message embedding, we design a multi-mask mechanism to identify robust regions in the latent space, aiming to enhance the resilience of the steganographic method against operations such as video compression.

**Invariance Mask $M_1$:** This mask highlights latent positions that remain stable after encoder-decoder and noise addition. As shown in the "Preprocess" section of Fig. 2, we apply encoding, decoding, and noise to $X_0$ to simulate the distortion that may occur when the generated video is used in real-world scenarios. We use $Noise(\cdot)$ to denote lossy processing such as video compression. The above process can be expressed as:

$$X_0' = E(Noise(D(X_0))). \quad (7)$$

We then compute the $L_1$ distance between $X_0$ and $X_0'$ as:

$$d_1(c, f, h, w) = \|X_0(c, f, h, w) - X_0'(c, f, h, w)\|, \quad (8)$$

where $(c, f, h, w)$ indices valid positions within the spatial-temporal dimensions.

The invariance mask $M_1$ is defined as the set of position indices corresponding to the smallest $\tau_1\%$ values in $d_1$. We define this function as $Mask(\cdot)$, formally expressed as:

$$M_1(c, f, h, w) = Mask(d_1, \tau_1) = \begin{cases} 1 & \text{if } d_1(c, f, h, w) \in \text{top smallest } \tau_1, \\ 0 & \text{otherwise.} \end{cases} \quad (9)$$

$M_1 = 1$ marks invariant regions resilient to codec distortions and noise.

**Discriminative Mask $M_2$:** This mask identifies latent positions where $X_0^1$ and $X_0^2$ differ the most, enabling reliable bit discrimination. Similarly, we compute $L_1$ distances between $X_0^1$ and $X_0^2$:

$$d_2(c, f, h, w) = \|X_0^1(c, f, h, w) - X_0^2(c, f, h, w)\|. \quad (10)$$

The discriminative mask $M_2$ is defined as the set of position indices corresponding to the largest $\tau_2\%$ values in $d_2$, formally expressed as:

$$M_2(c, f, h, w) = I - Mask(d_2, 1 - \tau_2) = \begin{cases} 1 & \text{if } d_2(c, f, h, w) \in \text{top largest } \tau_2, \\ 0 & \text{otherwise,} \end{cases} \quad (11)$$

where $I$ is the identity matrix.

**Combined Mask $M$:** The final mask is the dot product of $M_1$ and $M_2$, retaining positions that are both invariant ($M_1 = 1$) and discriminative ($M_2 = 1$):

$$M = M_1 \odot M_2, M \in \mathbb{R}^{C' \times F' \times H' \times W'}. \tag{12}$$

Positions where $M = 1$ (both masks are 1) are used for message embedding, ensuring robustness.

### 3.2.3 Embedding

On the basis of the analysis in Section 3.2.2, the steganographic capacity of LD-RoViS depends on $\tau_1$ and $\tau_2$. For a generated video $V_s$, assume that $M_1$ and $M_2$ are independent, the embedding capacity $n$ is given by:

$$n = C' \times F' \times H' \times W' \times \tau_1 \times \tau_2, \tag{13}$$

where slight variations may occur due to ties in percentile rankings. Accordingly, we embed a binary message $m$ of length $n$, and construct an embedding matrix $H$ from $m$ via the following rule:

$$H(c, f, h, w) = Transform(M, m) = \begin{cases} -1 & \text{if } M(c, f, h, w) = 0, \quad \text{(non-embedding region)} \\ m_k & \text{if } M(c, f, h, w) = 1, \quad \text{(embedding region)} \end{cases} \tag{14}$$

where $m_k \in \{0, 1\}$ is the $k$-th bit of $m$, which is filled row-wise from top-left to bottom-right.

The stego latent $X_s$ is formed by mixing $X_0$, $X_0^1$, and $X_0^2$ according to $H$:

$$X_s(c, f, h, w) = Embedding(X_0, X_0^1, X_0^2, H) = \begin{cases} X_0(c, f, h, w) & \text{if } H(c, f, h, w) = -1, \\ X_0^1(c, f, h, w) & \text{if } H(c, f, h, w) = 0, \\ X_0^2(c, f, h, w) & \text{if } H(c, f, h, w) = 1. \end{cases} \tag{15}$$

Finally, $X_s$ is decoded to the stego video $V_s = D(X_s)$ for transmission.

---

**Algorithm 1** Message Embedding Algorithm

1: **Input:** $prompt$, $seed$, $k_s$, cfg, message $m$, thresholds $\tau_1$, $\tau_2$
2: **Output:** stego video $V_s$
3: **Shared:** $G$, $D$, $E$
4:     Sample $X_T$ from $\mathcal{N}(0, \mathbf{I})$
5:     Perform $T$-step denoising:
6:     $X_0 \leftarrow Diffuse(G, X_T, k_s)$
7:     $X_0^1 \leftarrow Diffuse(G, X_1, k_s + \text{cfg})$
8:     $X_0^2 \leftarrow Diffuse(G, X_1, k_s - \text{cfg})$
9: **Preprocess:**
10:     $X_0' \leftarrow E(Noise(D(X_0)))$
11:     $M_1 \leftarrow Mask(L_1(X_0, X_0'), \tau_1)$
12:     $M_2 \leftarrow I - Mask(L_1(X_0^1, X_0^2), 1 - \tau_2)$
13:     $M \leftarrow M_1 \odot M_2$
14: **Message Encoding:**
15:     $H \leftarrow Transform(M, m)$
16: **Latent Mixing:**
17:     For each $(c, f, h, w)$ in latent space:
18:       **if** $H(c, f, h, w) = -1$ **then** $X_s(c, f, h, w) \leftarrow X_0(c, f, h, w)$
19:       **elif** $H(c, f, h, w) = 0$ **then** $X_s(c, f, h, w) \leftarrow X_0^1(c, f, h, w)$
20:       **else** $X_s(c, f, h, w) \leftarrow X_0^2(c, f, h, w)$
21:     $V_s \leftarrow D(X_s)$
22: **return** $V_s$

**Algorithm 2** Message Extraction Algorithm

1: **Input:** $prompt$, $seed$, $k_s$, cfg, received video $V_s'$, thresholds $\tau_1$, $\tau_2$
2: **Output:** recovered message $m'$
3: **Shared:** $G$, $D$, $E$
4:     Sample $X_T$ from $\mathcal{N}(0, \mathbf{I})$
5:     Perform $T$-step denoising:
6:     $X_0 \leftarrow Diffuse(G, X_T, k_s)$
7:     $X_0^1 \leftarrow Diffuse(G, X_1, k_s + \text{cfg})$
8:     $X_0^2 \leftarrow Diffuse(G, X_1, k_s - \text{cfg})$
9: **Preprocess:**
10:     $X_0' \leftarrow E(Noise(D(X_0)))$
11:     $M_1 \leftarrow Mask(L_1(X_0, X_0'), \tau_1)$
12:     $M_2 \leftarrow I - Mask(L_1(X_0^1, X_0^2), 1 - \tau_2)$
13:     $M \leftarrow M_1 \odot M_2$
14: **Stego Latent Extraction:**
15:     $X_s' \leftarrow E(V_s')$
16: **Message Decoding:**
17:     Initialize empty message $m'$
18:     For each $(c, f, h, w)$ where $M(c, f, h, w) = 1$:
19:       $d_1 \leftarrow \|X_s'(i, j) - X_0^1(i, j)\|$
20:       $d_2 \leftarrow \|X_s'(i, j) - X_0^2(i, j)\|$
21:       **if** $d_1 < d_2$ **then** append 0 to $m'$
22:       **else** append 1 to $m'$
23: **return** $m'$

## 3.3 Message Extraction

The receiver, equipped with the shared $prompt$ and parameters $(seed, k_s, \text{cfg})$, regenerates $X_0, X_0^1, X_0^2$, and $M$ via the same steps ("Shared" section in Fig. 2). The received video $V_s'$ is encoded to $X_s' = E(V_s')$.

For each position $(c, f, h, w)$ where $M(c, f, h, w) = 1$, the receiver compares the $L_1$ distances between $X_s'$ and $X_0^1, X_0^2$ to recover the message $m' = Extraction(X_s', X_0^1, X_0^2, M)$:

$$d_1'(c, f, h, w) = \|X_s'(c, f, h, w) - X_0^1(c, f, h, w)\|, \tag{16}$$

$$d_2'(c, f, h, w) = \|X_s'(c, f, h, w) - X_0^2(c, f, h, w)\|, \tag{17}$$

$$m_k' = \begin{cases} 0 & \text{if } d_1'(c, f, h, w) < d_2'(c, f, h, w) \text{ and } M(c, f, h, w) = 1, \\ 1 & \text{if } d_1'(c, f, h, w) \geq d_2'(c, f, h, w) \text{ and } M(c, f, h, w) = 1, \end{cases} \tag{18}$$

where $m_k' \in \{0, 1\}$ is the $k$-th bit of $m'$, which is filled row-wise from top-left to bottom-right.

The embedding and extraction procedures are shown in Algorithm 1 and Algorithm 2, respectively. According to Equations 16, 17, and 18, the accuracy of message extraction is highly dependent on the difference between $X_0^1$ and $X_0^2$. To simplify the design while maintaining this distinguishability, we generate the two variables using symmetric CFG scales, $i.e.$, $k_s + \text{cfg}$ and $k_s - \text{cfg}$, where cfg is a tunable hyperparameter controlling the modulation intensity.

# 4 Experiment

## 4.1 Experiment Setup

Our proposed LD-RoViS is built upon a deterministic latent video diffusion model. To this end, we adopt the T2V-1.3B model from Wan2.1 [30] as our video model. This model leverages Flow Matching sampling to efficiently produce high-quality videos. For evaluation, we use VidProM [47], a large-scale and diverse text-to-video prompt dataset. We randomly sample 100 prompts from VidProM and generate corresponding videos via Wan2.1. Each video has a resolution of 480×832, a duration of 5 seconds, and a frame rate of 16 fps, resulting in 81 frames per video. All subsequent experiments are conducted on these 100 prompts with $seed = 99, k_s = 5.0$ and run on four NVIDIA RTX A6000 GPUs, each with 48 GB of VRAM. In our experiments, we set the hyperparameters as $\tau_1 = 0.32$, $\tau_2 = 0.02$, and $cfg = 16$. Additional experiments and analysis of these hyperparameters can be found in the Appendix A, including experiments on a non-deterministic model (LTX-Video [48]).

## 4.2 Evaluation Metrics

We evaluate the performance of our steganographic method via the following metrics:

**Accuracy** (acc, %) is defined as the ratio of correctly recovered bits in $m'$ to $m$. A higher acc indicates greater robustness.

**Peak Signal-to-Noise Ratio** (PSNR, dB) measures the perceptual similarity between two images; a higher PSNR indicates better image quality. In this work, the PSNR is computed as the average PSNR across all frames between $V_c$ and $V_s$.

**BRISQUE** [49] is a no-reference image quality assessment metric commonly used to evaluate the perceptual quality of generated images. Lower BRISQUE scores correspond to higher perceptual quality. Here, we define it as the average BRISQUE score over all frames of $V_s$.

**Capacity** refers to the total number of data bits that can be embedded in the stego video $V_s$.

**Error Rate** ($P_E$, %) denotes the error rate in steganalysis detection. A higher $P_E$ indicates better security of the steganographic method.

## 4.3 Quantitative Performance Evaluation

### 4.3.1 Metric Experiments

To evaluate the performance of the proposed method, we conducted experiments on 100 videos generated from diverse prompts. As shown in Table 1, we report four metrics: PSNR, BRISQUE

score, accuracy (acc), and embedding capacity. Each result is presented as the mean and standard deviation over the 100 test videos. We compare our method with three recent video steganography methods: two transform-domain methods (AQIM [12] and MEC_AQIM [13]), and a face-swapping-based generative method (RoGVSN [42]). For a fair comparison, we set the embedding capacity of AQIM and MEC_AQIM to 10,000 bits and tested them under the same conditions. Since RoGVSN only supports face-related video and does not allow for adjustable capacity, we selected 10 face-related videos from the test set for its evaluation. Additionally, the PSNR is not applicable to RoGVSN because of its face-swapping nature, and the corresponding values are left blank. The experimental results show that our method significantly outperforms the baselines in terms of visual quality, while maintaining high extraction accuracy and embedding capacity.

Table 1: Comparison of performance. The results are presented as the means and standard deviations.

| Method | PSNR↑ | BRISQUE↓ | acc (%)↑ | capacity↑ |
|---|---|---|---|---|
| AQIM | $34.81 \pm 0.44$ | $32.87 \pm 6.06$ | $\mathbf{99.44 \pm 0.27}$ | 10000 (fixed) |
| MEC_AQIM | $35.21 \pm 0.47$ | $32.71 \pm 6.10$ | $90.99 \pm 5.90$ | 10000 (fixed) |
| RoGVSN | – | $49.53 \pm 4.55$ | $99.28 \pm 0.38$ | 729 (fixed) |
| Ours | $\mathbf{41.66 \pm 1.52}$ | $\mathbf{28.90 \pm 6.05}$ | $99.17 \pm 0.63$ | $\mathbf{11983 \pm 1446}$ |

#### 4.3.2 Security Experiments

To evaluate the resistance of our proposed LD-RoViS against steganalysis attacks, we compare it with three baseline methods. Following the experimental setup in [12], we adopt two image steganalyzers, CovNet [24] and LWENet [25], and one video steganalysis feature extractor, SUPERB [50] combined with a linear classifier [51]. We test 100 generated cover-stego video pairs. For a fair comparison, the capacity of AQIM and MEC_AQIM is set to 10,000 bits. Since the capacity of RoGVSN is fixed, we follow its original design with a payload of 729 bits. Using ffmpeg, we decode these videos to obtain 8,100 pairs of cover and stego frames, with 4,000 used for training, 600 for validation, and 3,500 for testing. As shown in Table 2, LD-RoViS effectively resists steganalysis attacks, achieving error rates close to 50% (random guessing).

Table 2: $P_E$ (%,↑) of steganalysis.

| Method | SUPERB | CovNet | LWENet |
|---|---|---|---|
| AQIM | 49.14 | 0.13 | 0.26 |
| MEC_AQIM | 47.32 | 0.01 | 1.07 |
| RoGVSN | 47.58 | 0.36 | 2.61 |
| ours | **49.18** | **49.74** | **48.49** |

Table 3: acc(%) under different compression and noise.

| Method | - | CRF=18 | CRF=23 | CRF=27 | noise | salt&pepper | brightness |
|---|---|---|---|---|---|---|---|
| AQIM | **99.44** | 91.24 | 90.67 | 87.49 | 82.46 | 80.04 | 48.93 |
| MEC_AQIM | 90.99 | 82.83 | 82.29 | 78.87 | 72.83 | 71.60 | 50.31 |
| RoGVSN | 99.28 | **97.42** | **97.06** | **97.04** | **96.20** | 94.45 | 96.05 |
| ours | 99.17 | 95.89 | 93.70 | 91.67 | 92.82 | **98.72** | **99.02** |

#### 4.3.3 Robustness Experiments

To evaluate the robustness of LD-RoViS against video compression and noise, we conducted the following experiments. The most common processing method on social media platforms is H.264 compression, where the Constant Rate Factor (CRF) is a key quality control parameter, typically set to 23. In our experiments, we tested both CRF=18 and CRF=27. Additionally, we applied common noise perturbations: "noise" refers to Gaussian noise with a standard deviation of 0.05, "salt&pepper[3]" denotes impulse noise with a probability of 0.01, and "brightness" represents an increase of 0.1 in the HSV color space. As shown in Table 3, LD-RoViS demonstrates strong robustness to brightness changes and salt-and-pepper noise, while maintaining over 90% extraction accuracy under other lossy conditions. Although RoGVSN is more robust against H.264 compression and Gaussian noise, it is important to note that its embedding capacity is only 729 bits, whereas LD-RoViS supports approximately 12,000 bits.

### 4.4 Subjective performance evaluation

To evaluate the impact of steganographic embedding on video visual quality, we present several visual examples in Fig. 3. Using FFmpeg, we decode both cover and stego videos and extract the middle

---

[3]Salt-and-pepper noise is applied in an image-level manner, where a single spatial impulse mask is shared across frames with an overall probability of 0.01.

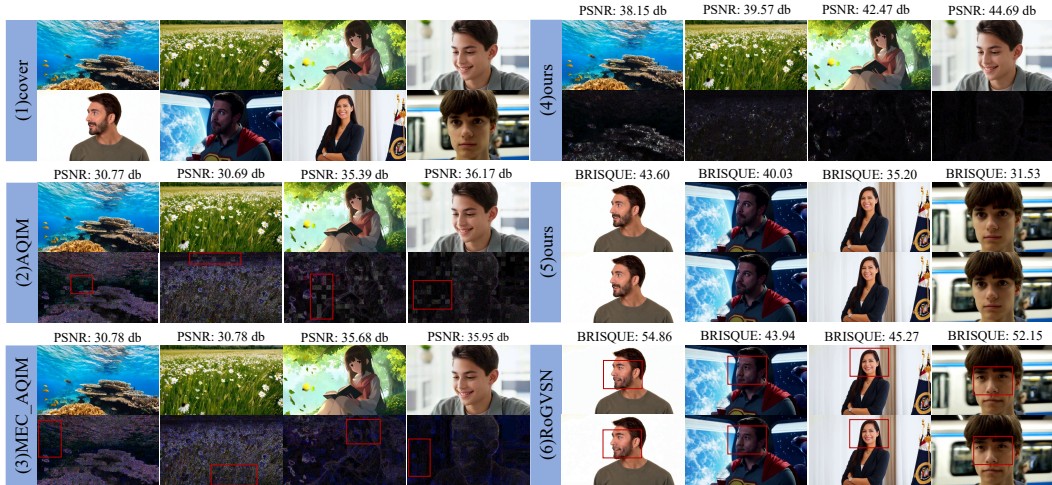

Figure 3: Visual comparisons between our method and baselines. In (2)–(4), the second row shows the pixel differences between the cover and stego frames (brightness increased by 10×), whereas in (5)–(6), the second row shows the temporally adjacent frames.

frame (the 41st frame). In SubFig. 3 (2), (3), and (4), the top row shows the stego frames, whereas the bottom row shows the pixel-wise differences between the stego and cover frames (brightness enhanced 10× for better visibility). As shown, traditional transform-domain methods are sensitive to video encoding and introduce noticeable block artifacts. In contrast, our method avoids direct modification of video content, resulting in only minor pixel differences in texture-rich regions. In SubFig. 3 (5) and (6), the top row shows the stego frame, whereas the bottom row displays the subsequent frame in temporal order (i.e., the 42nd frame). Since RoGVSN embeds messages via face-swapping, noticeable blur and distortion are observed in facial regions, leading to lower perceptual quality. These results demonstrate that our method balances effective steganography with minimal perceptual impact, outperforming baselines in preserving video quality for real-world applications.

## 4.5 Ablation Studies

To further investigate the effectiveness of the multi-mask mechanism, we conducted ablation studies on the two masks $M_1$ and $M_2$. Specifically, we evaluated three different variants of the full model, with their differences summarized in Table 4. Each variant was tested on 10 generated videos, and the average results are reported in Table 5. The experimental results show that removing either $M_1$ or $M_2$ leads to a significant decrease in accuracy, confirming the effectiveness of $M_1$ in identifying invariant regions and $M_2$ in identifying discriminative regions within the latent space. Their combined effect enables the identification of robust areas in the latent space to achieve robust steganography.

Table 4: Ablation variants.

| Method | Mask $M_1$ | Mask $M_2$ |
|---|---|---|
| variant#1 | × | × |
| variant#2 | ✓ | × |
| variant#3 | × | ✓ |
| ours | ✓ | ✓ |

Table 5: Performance of different variants.

| Method | acc(%)↑ | PSNR(db)↑ | BRISQUE↓ | capacity(bits)↑ |
|---|---|---|---|---|
| variant#1 | 62.67 | 35.39 | 30.55 | **1935111** |
| variant#2 | 75.46 | 37.49 | 29.47 | 617913 |
| variant#3 | 88.59 | 40.53 | 29.01 | 41132 |
| ours | **99.17** | **41.66** | **28.90** | 11983 |

## 5 Conclusion

In this work, we present a robust video steganography method for deterministic latent diffusion models. Our method innovatively constructs a steganographic channel by leveraging the classifier-free guidance (CFG) scale of diffusion models. In addition, we introduce a multi-mask mechanism based on adversarial encoding-decoding and video compression perturbations to identify invariant and distinguishable regions in the latent space, thereby ensuring robustness. To the best of our knowledge,

this is the first steganographic method for deterministic latent diffusion, which achieves significant advantages in both embedding capacity and security compared with existing video steganography methods.

# 6    Acknowledgements

This work was supported in part by the National Natural Science Foundation of China under Grant U2336206, Grant 62472398, Grant U2436601, and Grant 62402469.

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

# A  Hyperparameter Tuning

In the LD-RoViS framework, the optimal values of three hyperparameters—$\tau_1$, $\tau_2$, and *cfg*—need to be determined through experiments. We generated stego videos using 10 different prompts and computed the average values of accuracy (acc), PSNR, BRISQUE, and capacity. The results are summarized in Table 6, 7, 8. To evaluate $\tau_1$, we fixed $\tau_2 = 0.02$ and *cfg* = 16; to evaluate $\tau_2$, we fixed $\tau_1 = 0.32$ and *cfg* = 16; and to evaluate *cfg*, we fixed $\tau_1 = 0.32$ and $\tau_2 = 0.02$.

We observe that as $\tau_1$ and $\tau_2$ increase, capacity increases, whereas acc, PSNR, and BRISQUE decrease. Similarly, as *cfg* increases, acc improves, PSNR and BRISQUE decrease, and capacity remains nearly unchanged. Our goal is to maximize capacity while maintaining acc $\geq$ 99%. On this basis, we choose the combination $\tau_1 = 0.32$, $\tau_2 = 0.02$, and *cfg* = 16. Notably, the BRISQUE score of the cover video is 28.83, which is very close to that of our stego videos. This finding indicates that although there are content differences between the stego and cover videos, their perceptual quality, as measured without reference, remains similar.

Table 6: Performance under different values of $\tau_1$.

| $\tau_1$ | acc(%)↑ | PSNR(db)↑ | BRISQUE↓ | capacity(bits)↑ |
|---|---|---|---|---|
| 0.01 | 99.45 | **43.11** | **28.80** | 364 |
| 0.02 | 99.31 | 42.99 | 28.84 | 721 |
| 0.04 | **99.52** | 42.83 | 28.84 | 1464 |
| 0.08 | 99.42 | 42.56 | 28.83 | 2913 |
| 0.16 | 99.42 | 42.18 | 28.91 | 5869 |
| 0.32 | 99.17 | 41.66 | 28.90 | 11983 |
| 0.64 | 97.84 | 40.97 | 28.94 | **24995** |

Table 7: Performance under different values of $\tau_2$.

| $\tau_2$ | acc(%)↑ | PSNR(db)↑ | BRISQUE↓ | capacity(bits)↑ |
|---|---|---|---|---|
| 0.01 | **99.65** | **42.18** | **28.89** | 5957 |
| 0.02 | 99.17 | 41.66 | 28.90 | 11983 |
| 0.04 | 98.53 | 41.01 | 28.96 | 23511 |
| 0.08 | 97.25 | 40.10 | 29.05 | 47708 |
| 0.16 | 94.80 | 39.02 | 29.25 | 100786 |
| 0.32 | 90.85 | 38.12 | 29.36 | 204572 |
| 0.64 | 82.90 | 37.57 | 29.48 | **413237** |

Table 8: Performance under different values of *cfg*.

| *cfg* | acc(%)↑ | PSNR(db)↑ | BRISQUE↓ | capacity(bits)↑ |
|---|---|---|---|---|
| 1 | 60.72 | **43.15** | **28.83** | 12168 |
| 2 | 68.56 | 43.04 | 28.84 | **12206** |
| 4 | 83.72 | 42.84 | 28.84 | 12083 |
| 8 | 95.45 | 42.43 | 28.79 | 12105 |
| 16 | **99.17** | 41.66 | 28.90 | 11983 |

# B  Validation on Other Video Diffusion Models

To further evaluate the generalization ability of the proposed LD-RoViS, we additionally implemented the method on another open-source text-to-video generation model, LTX-Video [48] (ltxv-2b-0.9.8-distilled version). LTX-Video is a two-stage diffusion model. The first stage focuses on generating high-quality image frames, while the second stage refines temporal consistency across frames. It

adopts a DiT-based architecture and employs an EDM sampler (non-deterministic). In our implementation, we retained the first stage unchanged and inserted the steganographic module at the final step of the second stage's denoising process. Due to architectural and sampling differences between Wan2.1 and LTX-Video, we re-tuned the hyperparameters using a small validation set and obtained the following optimal values:

$$k_s = 1.1, \quad \text{cfg} = 1.8, \quad \tau_1 = 0.32, \quad \tau_2 = 0.02.$$

Note that LTX-Video's generated video quality is slightly inferior to that of Wan2.1, which explains the higher BRISQUE scores reported in Table 9. In addition, the latent size in LTX-Video is approximately 2/5 of Wan2.1's latent size, leading to a corresponding reduction in embedding capacity. Experimental results demonstrate that LD-RoViS maintains stable performance under this different architecture and sampler, confirming the adaptability of our method.

Table 9: Performance comparison of LD-RoViS on different video generation models.

| Model | Accuracy (%)↑ | PSNR (dB)↑ | BRISQUE ↓ | Capacity (bits)↑ |
|---|---|---|---|---|
| LD-RoViS (Wan2.1) | 99.17 | 41.66 | 28.90 | 11983 |
| LD-RoViS (LTX-Video) | 99.23 | 41.39 | 34.21 | 4281 |

## C  Ablation on Modulation Time Step

To investigate the effect of the modulation time step in the CFG modulation strategy, we performed ablation studies by applying the modulation at different denoising steps. Here, $t$ denotes the reverse denoising step counting from the last step, *i.e.*, $t = 2$ corresponds to the second-to-last denoising step. The results are shown in Table 10. When the modulation is applied at earlier denoising steps, the resulting latent variables $X_0^1$ and $X_0^2$ become more uncontrollable. After several denoising steps, the difference between $X_0^1$ and $X_0^2$ may either increase or decrease unpredictably. As our message extraction relies heavily on the distinguishability between $X_0^1$ and $X_0^2$, this uncertainty leads to lower extraction accuracy with higher standard deviations. Meanwhile, PSNR, BRISQUE, and steganalysis results remain nearly unchanged, indicating that perceptual quality and security are preserved. Therefore, the final-step modulation is adopted in our design.

Table 10: Ablation study on different modulation time steps.

| Time step $t$ | Accuracy (%)↑ | PSNR (dB)↑ | BRISQUE ↓ | CovNet $P_e$(%)↑ |
|---|---|---|---|---|
| $t=1$ (Ours) | 99.17±0.63 | 41.66 | 28.90 | 49.74 |
| $t=2$ | 98.43±1.12 | 41.22 | 29.99 | 49.93 |
| $t=4$ | 95.72±2.57 | 41.34 | 28.06 | 49.78 |
| $t=8$ | 94.82±3.30 | 41.94 | 29.62 | 49.72 |

## D  Time Consumption

The embedding time is also a critical metric for evaluating steganographic methods. Methods with lower embedding times are generally more practical for real-world applications. To assess this, we measured the embedding latency using Python's `time` package, and the results are shown in Table 11. Notably, since LD-RoViS performs steganographic embedding during the video generation process, the actual embedding time should be calculated by subtracting the time required to generate a clean video via the video model. The experimental results show that LD-RoViS introduces minimal time overhead (30.48 s), significantly outperforms AQIM and MEC_AQIM, and achieves a runtime comparable to that of RoGVSN.

## E  More Experimental Details

In this section, we provide more experimental details to ensure the reproducibility of LD-RoViS. We adopt the T2V-1.3B model from Wan2.1 [30] as the video generation model. This model has

Table 11: Embedding latency (in seconds) for different methods.

| Method | Video Model | LD-RoViS | RoGVSN | AQIM | MEC_AQIM |
|---|---|---|---|---|---|
| Additional Time (s)↓ | 366.02 | 30.48 | 36.52 | 698.80 | 697.59 |

low memory requirements and can run on a single RTX 4090 Ti. It also offers fast generation speed, producing a video in just 50 time steps. On average, it takes approximately 6 minutes to generate an 81-frame video at a resolution of 480×832. For steganographic tasks, the average generation time is approximately 7 minutes per video.

**Dataset**: Our experiments use a dataset consisting of 100 cover videos and 100 stego videos. The prompts for the cover and stego videos are exactly the same. The cover videos are generated via a clean version of the Wan2.1 model, whereas the stego videos are produced via a Wan2.1 model with the steganographic module. All prompts are randomly selected from the VidProM dataset [47]. However, some prompts from VidProM contain parameters unsupported by Wan2.1. We manually filtered out such prompts. Table 12 lists a subset of the prompts used in our experiments.

Table 12: Example prompts used for video generation.

| 1 | a kangaroo dancing to electronic music in a crowded nightclub. |
|---|---|
| 2 | a lady wandering in the enchanted woods, lost and confused. |
| 3 | aliens walking in city. |
| 4 | cars on the beach are attacked by sharks. |
| 5 | Animals working together, sharing resources, and demonstrating cooperation under the guidance of Leo, the wise lion. |
| 6 | The black Chinese dragon opens its eyes. |
| 7 | a wolf walking in the jungle. |
| 8 | A massive storm hitting a city. |
| 9 | a wild cat running in the jungle. |
| 10 | an elephant riding a bike, masterpiece, cinematic. |

## F   More Visual Results

In this section, we present more visual results to evaluate the impact of steganographic embedding on video quality.

**Visual Quality across Video Frames**: A common challenge in traditional video steganography is distortion drift, where embedding distortions gradually propagate across frames, resulting in noticeable artifacts in later parts of the video. To investigate whether LD-RoViS suffers from a similar issue, we decomposed four generated stego videos into individual frames via FFmpeg. We then visualized the first frame, middle frame (41st), and last frame (81st) in Fig. 4. The results show that the steganographic modifications made in the latent space by LD-RoViS do not introduce visible distortions or artifacts. The visual quality remains stable throughout the video.

**Pixel-Wise Differences between Cover and Stego Videos**: In the main paper, Fig. 3 shows that the pixel differences between cover and stego videos generated by LD-RoViS are minimal and are mostly concentrated in regions with complex textures. In this section, we provide a deeper analysis of these results. Fig. 5 presents additional examples of pixel-level differences. The "diff" images represent the absolute pixel difference between the 41st frame of the cover and stego videos, with brightness amplified by a factor of 10 for better visibility. These visualizations confirm that pixel differences are extremely subtle and imperceptible to the human eye. Notably, most differences appear along object contours. We hypothesize that this is due to the low weighting of the predicted noise at the final step of the diffusion model's reverse process. Even large modifications to this predicted noise have minimal effects on the final output. This suggests that the last step of the diffusion process provides

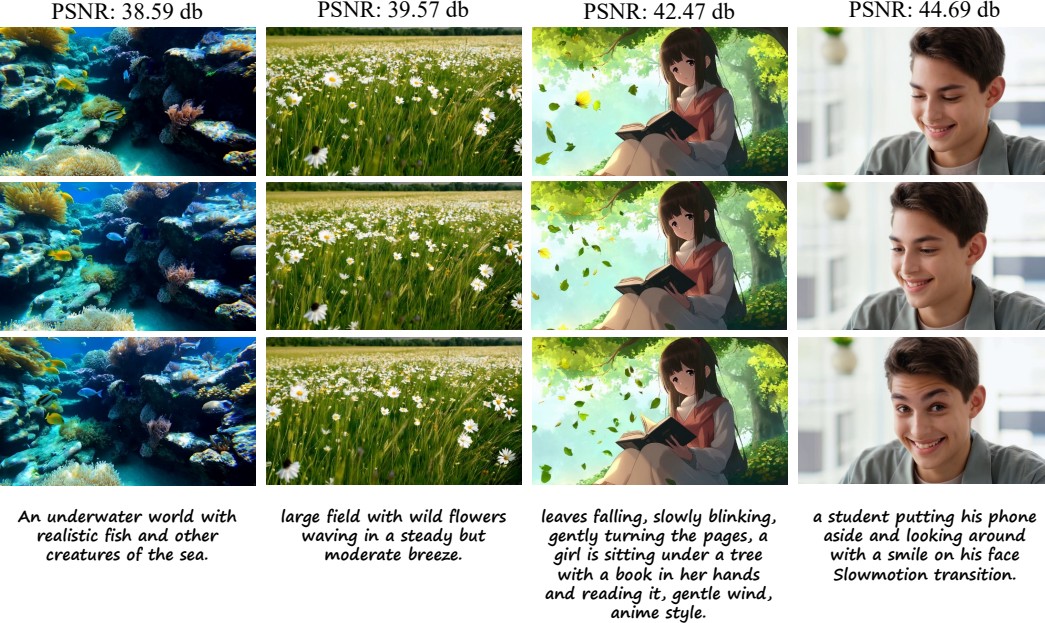

PSNR: 38.59 db     PSNR: 39.57 db     PSNR: 42.47 db     PSNR: 44.69 db

*An underwater world with realistic fish and other creatures of the sea.*    *large field with wild flowers waving in a steady but moderate breeze.*    *leaves falling, slowly blinking, gently turning the pages, a girl is sitting under a tree with a book in her hands and reading it, gentle wind, anime style.*    *a student putting his phone aside and looking around with a smile on his face Slowmotion transition.*

Figure 4: The first frame, middle frame (41st), and last frame (81st) of a stego video generated by LD-RoViS, along with their corresponding prompts.

a naturally robust channel for steganography—allowing hidden information to be embedded with negligible visual degradation and strong resistance to detection.

## G  Limitations

Although the proposed LD-RoViS enables robust video steganography and outperforms existing methods in terms of both security and visual quality, it also has certain limitations.

**The quality of the stego video depends heavily on the performance of the video generation model.** While video generation models are rapidly improving, they still occasionally produce low-quality outputs. As shown in Fig 6, some generated videos exhibit physically implausible deformations, artifacts, or distorted faces. These low-quality samples may raise suspicion during transmission between the sender and receiver, thereby reducing the practical usability of the steganographic method.

**LD-RoViS inevitably alters the video generation process.** Specifically, it modifies the parameters at the final timestep of the reverse diffusion process, which may introduce slight shifts in the distribution of the generated videos and lead to a decrease in visual quality. However, the experimental results show that steganalysis tools fail to detect meaningful distribution differences between the cover and stego videos, indicating that LD-RoViS remains secure against detection. Additionally, the stego videos maintain high PSNR and favorable BRISQUE scores, suggesting that LD-RoViS achieves strong practicality despite the minimal distortion introduced.

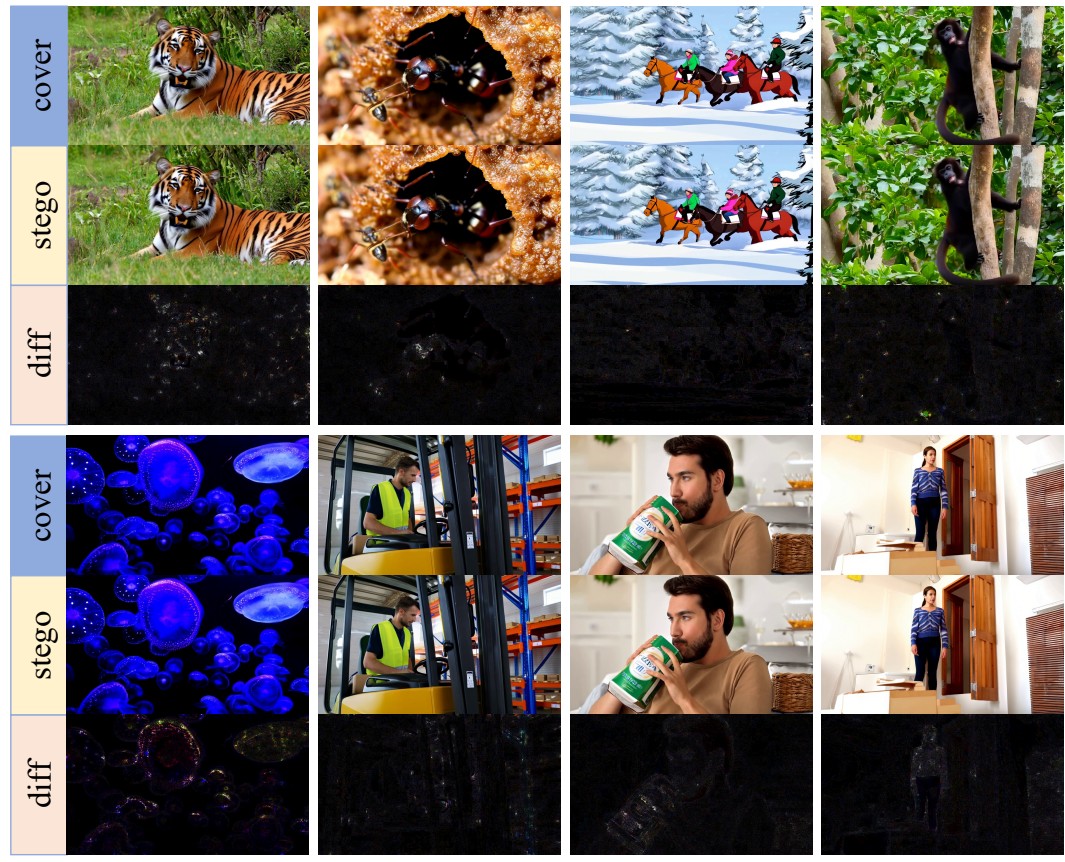

Figure 5: Cover video, stego video, and their pixel-wise difference ("diff" denotes the pixel difference, with brightness amplified by a factor of 10 for better visibility).

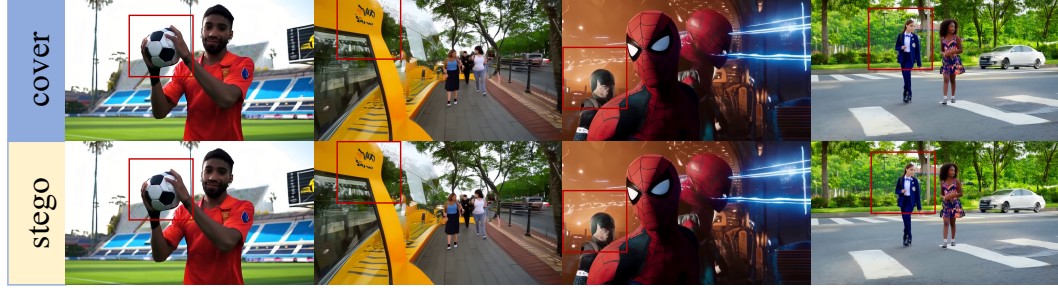

Figure 6: Examples of low-quality generated videos, where noticeable distortions and artifacts are present in the frames.

