# OpenReview forum: "LD-RoViS: Training-free Robust Video Steganography for Deterministic Latent Diffusion Model"
_NeurIPS.cc/2025/Conference — NeurIPS 2025 poster_

### Official Review · Reviewer_66yD · 2025-06-30

**Clarity:** 3
**Significance:** 3
**Originality:** 3
**Rating:** 5
**Confidence:** 2

**Summary:**

This paper proposes a generative steganography method utilizing pre-trained video diffusion models. It demonstrates high robustness against information loss during transmission by employing a classifier-free guidance trick and an embedding selection strategy called the Multi-mask mechanism.

**Questions:**

- In this method, can sender and receiver use M_2 = Mask( X_0, X_0^1), thereby avoiding the need to share k_2? Is there a specific reason for requiring k_2?

- Since the multi-mask mechanism enhances robustness, is it possible to increase capacity by embedding multi-bit messages using multiple values of k?

**Ethical Concerns:**

["NO or VERY MINOR ethics concerns only"]

**Final Justification:**

The rebuttal addresses previous concerns. I will keep my rating as acceptance.

**Limitations:**

Yes

**Quality:**

3

**Strengths And Weaknesses:**

[Strength]

- The novel multi-mask mechanism effectively addresses robustness issues commonly encountered in diffusion-based steganography methods, particularly those caused by encoder-decoder errors in latent-based diffusion models.

- The writing and figures in the paper are sufficiently clear to understand the proposed method.

[Weakness]

- The number of hidden messages is significantly reduced due to masking, particularly with M_2. It appears that using random noise instead of X_0^1 and X_0^2, as done in StegaDDPM, may alleviate this issue.

- It would be helpful to quantify the computational overhead compared to standard text-to-video generation using metrics like latency or GFLOPS.

- The proposed method is only validated on the T2V-1.3B model of WAN2.1. Readers may wonder whether it would also work on other widely-used open-source models such as LTX-Video [A].

[A] Lightricks, https://huggingface.co/Lightricks/LTX-Video

---

> ### Author Rebuttal · Authors · 2025-07-30
>
> We are pleased to receive your positive feedback. Thank you for taking the time to review our work and for recognizing our contributions to robustness. Below, we provide detailed responses to the concerns you have raised.
>
> `Weakness 1：The number of hidden messages is significantly reduced due to masking, particularly with M_2. It appears that using random noise instead of X_0^1 and X_0^2, as done in StegaDDPM, may alleviate this issue.`
>
> In fact, StegaDDPM and methods [1-3] cannot be applied to deterministic diffusion models, as explained below:
>
> The diffusion denoising sampling process can be simplified as:
>
> $X_s = \mu_\theta(X_t,t)+\sigma_s\epsilon, \epsilon\in N(0,I),s<t$
>
> When $\sigma_s = 0$, the sampling is deterministic. Mainstream video generation models (e.g., our used Wan2.1) adopt deterministic sampling to reduce computational cost. The secret message in StegaDDPM and related works [1-3] is embedded into **the random noise $\epsilon$ at the final denoising step, thus these methods cannot be applied to deterministic sampling models**.
>
> In contrast, our method only relies on modifying the CFG scale parameter at the final denoising step, i.e., modulating the $\mu_\theta(X_t, t)$ term. Therefore, our method supports both non-deterministic and deterministic sampling, which is a core innovation of our work.
>
> `Weakness 2：It would be helpful to quantify the computational overhead compared to standard text-to-video generation using metrics like latency or GFLOPS.`
>
> Thank you for your suggestion to quantify computational consumption. In the appendix, we previously measured time consumption only in seconds, which is indeed insufficient. Following your advice, we used Python’s `time` package to measure latency, and the results are shown in the table below:
>
> | Method | Video Model Inference Time  | ours  | RoGVSN  | AQIM  | MEC_AQIM  |
> | --- | --- | --- | --- | --- | --- |
> | Additional Time for embedding (s)↓ | 366.02 | 30.48 | 36.52 | 698.80 | 697.59 |
>
> Since our steganography method is training-free and plug-and-play, it does not modify the training parameters of the base generation model and only adds an extra stego module during inference, we did not use GFLOPS metrics for evaluation.
>
> `Weakness 3：The proposed method is only validated on the T2V-1.3B model of WAN2.1. Readers may wonder whether it would also work on other widely-used open-source models such as LTX-Video [4].`
>
> Thank you for suggesting LTX-Video as an open-source alternative. Your suggestion is very valuable. We implemented our steganography method on the `ltxv-2b-0.9.8-distilled` version of LTX-Video. We carefully studied its technical documentation and inference code, successfully integrated our method, and conducted performance tests. The results are shown below:
>
> | Method | Accuracy (%)↑ | PSNR (dB)↑ | BRISQUE ↓ | Capacity (bits)↑ |
> | --- | --- | --- | --- | --- |
> | ours (Wan2.1) | 99.17 | 41.66 | 28.90 | 11983 |
> | ours (LTX-Video) | 99.23 | 41.39 | 34.21 | 4281 |
>
> LTX-Video is a two-stage diffusion model. The first stage generates image frames with a focus on visual quality, while the second stage optimizes temporal coherence. It is based on a DiT architecture and uses a more advanced EDM sampler compared to DDIM. We kept the first stage unchanged and inserted our steganographic module at the final step of the second stage's denoising process. Due to differences between Wan2.1 and LTX-Video, we re-conducted hyperparameter experiments and determined the following hyperparameter values by testing on a small number of samples: $k_s=1.1, \text{cfg}=1.8, \tau_1=0.32, \tau_2=0.02.$ The results show that our method maintains performance even under this different architecture and sampler.
>
> Note that LTX-Video's video quality is slightly inferior to Wan2.1, which explains the higher BRISQUE scores of LTX-Video. Also, the latent size in LTX-Video is approximately 2/5 of Wan2.1’s latent size, leading to a reduction in capacity.
>
> Below are our responses to the questions you raised:
>
> `Question 1: In this method, can sender and receiver use M_2 = Mask( X_0, X_0^1), thereby avoiding the need to share k_2? Is there a specific reason for requiring k_2?`
>
> Indeed, our method can be simplified to avoid explicitly using $k_2$, requiring only $\text{cfg}$ and $k_s$. As mentioned in Section 3.3 of our paper, we set $k_1$ = $k_s$ - cfg and $k_2$ = $k_s$ + cfg, so both $k_1$ and $k_2$ are controlled by a single hyperparameter cfg. And in this case, $d_2 = ||X_0^1 - X_0^2|| = 2\times ||X_0 - X_0^1||$. Since the Discriminative Mask $M_2$ captures the $\tau_2$% most-different positions between $X_0^1$ and $X_0^2$, therefore $M_2 = \text{Mask}(X_0^1, X_0^2) = \text{Mask}(X_0, X_0^1)$.
>
>
> Since message extraction depends primarily on the difference between $X_0^1$ and $X_0^2$, and this difference is determined by the distance between $k_1$ and $k_2$, a single parameter cfg is sufficient. Considering that the design of $k_1$ and $k_2$ is somewhat redundant, we appreciate your suggestion and will remove their description in the final submission, retaining only the hyperparameter **cfg**. This simplification reduces the number of tunable hyperparameters, making the method more practical for real-world deployment.
>
> `Question 2: Since the multi-mask mechanism enhances robustness, is it possible to increase capacity by embedding multi-bit messages using multiple values of k?`
>
> Thank you for raising this insightful question. This idea has indeed been experimentally explored in prior work [2]. Specifically, during message extraction, if we use 4 different keys, we can obtain four corresponding variables $X_0^1, X_0^2, X_0^3, X_0^4$, thereby enabling us to encode 2-bit messages at each position and effectively doubling the embedding capacity.
>
> However, this introduces **a clear trade-off between capacity and robustness**. In practice, to ensure accurate extraction between just two variables $X_0^1$ and $X_0^2$, we already apply a strict filtering threshold ($\tau_2 = 0.02$ of Mask $M_2$), which filters about 98% of the regions. Introducing additional keys would further reduce the usable embedding area, potentially making message extraction unreliable. Hence, we did not pursue the multi-key scenario further.
>
> [1] Ldstega: Practical and robust generative image steganography based on latent diffusion models. *Proceedings of the 32nd ACM International Conference on Multimedia*. 2024.
>
> [2] PSyDUCK: Training-Free Steganography for Latent Diffusion. *arXiv preprint arXiv:2501.19172* (2025).
>
> [3] Pulsar: Secure steganography for diffusion models. *Proceedings of the 2024 on ACM SIGSAC Conference on Computer and Communications Security*. 2024.
>
> [4] Ltx-video: Realtime video latent diffusion. *arXiv preprint arXiv:2501.00103* (2024).

---

> > ### Comment · Reviewer_66yD · 2025-08-05
> >
> > Thank you for the detailed rebuttal. I hope this work to be accepted

---

> > > ### Author Response · Authors · 2025-08-05
> > >
> > > Thank you for your encouraging feedback and for taking the time to consider our rebuttal. We truly appreciate your support and constructive review.

---

> ### Comment · Area_Chair_PuqW · 2025-08-08
>
> Dear Reviewer,
>
> Could you give some feedback on the rebuttal provided by the authors?
>
> Your AC

---

### Official Review · Reviewer_4czU · 2025-07-03

**Clarity:** 3
**Significance:** 2
**Originality:** 3
**Rating:** 4
**Confidence:** 4

**Summary:**

This paper introduces LD-RoViS, a training-free and compression-resilient video steganography framework that embeds secret information into latent variables by modulating the final timestep of a deterministic diffusion process.

**Questions:**

To address concerns regarding generalizability and reliability,
- testing transferability to alternative generative backbones
- analyzing reconstruction failures to understand the inherent brittleness of the encoding scheme

**Ethical Concerns:**

["NO or VERY MINOR ethics concerns only"]

**Final Justification:**

After reviewing the authors’ rebuttal and taking into account the comments from other reviewers, I have decided to raise my score.

**Limitations:**

yes

**Paper Formatting Concerns:**

No formatting issues found.

**Quality:**

2

**Strengths And Weaknesses:**

Strengths
- The proposed multi-mask mechanism selectively identifies stable regions in latent space, leading to strong resilience against video compression artifacts, which are prevalent in social media platforms.
- By operating at the final timestep, the method maintains determinism and enables precise control over where and how secret content is embedded in the latent space.

Weaknesses
- The method relies heavily on deterministic reverse sampling, restricting its use to a narrow subset of diffusion models. As the community shifts toward stochastic samplers or improved generative backbones (e.g., DiT), the method’s relevance may diminish without adaptation.
- Despite focusing on robustness, the method does not always guarantee full secret message reconstruction, even under clean conditions. This raises concerns about its practical reliability in high-stakes environments.

---

> ### Author Rebuttal · Authors · 2025-07-30
>
> Thank you for your time and valuable suggestions. Below, we provide point-by-point responses to the concerns you raised.
>
> `Weakness 1：The method relies heavily on deterministic reverse sampling, restricting its use to a narrow subset of diffusion models. As the community shifts toward stochastic samplers or improved generative backbones (e.g., DiT), the method’s relevance may diminish without adaptation.`
>
> Thank you for your concern regarding the applicability of our method. First, we would like to clarify a misunderstanding about the scope of our method: our method is applicable to both **deterministic and non-deterministic sampling** processes. In contrast, existing diffusion-based image steganography methods [1-3] that do not rely on inversion typically support only non-deterministic samplers. We address this limitation through a **parameter modulation strategy** that only adjusts the CFG (Classifier-Free Guidance) scale during the denoising process. To the best of our knowledge, mainstream text-to-video generation models all support CFG scale, making our method broadly applicable.
>
>
>
> - **Our method is compatible with non-deterministic sampling and does not need reverse inversion:**
>
>
>
> Although our method, like [1–3], requires the sender and receiver to share the generation process, this can be easily achieved by **synchronizing the prompt and a random seed** (as illustrated by the `seed` in Figure 2). While non-deterministic samplers introduce random noise at each step, the noise sequence becomes deterministic when the random seed is fixed, and thus can be shared between both sender and receiver. Therefore, **our method remains applicable to non-deterministic sampling**, just as in methods [1]–[3].
>
>
> Furthermore, our method **does not require any inversion of the sampling**. The steganographic embedding is performed at the **final denoising step**, and the secret message can be accurately recovered simply by passing the stego video through the VAE encoder (the "$E$" module in Figure 2). This design does not rely on the reversibility of the sampler, making it compatible with a wide range of diffusion-based generative models. Notably, the official documentation of Wan2.1 states that "**Wan2.1 is designed on the mainstream diffusion transformer paradigm**," confirming that it adopts the DiT (Diffusion Transformer) architecture.
>
> - **Performance on another DiT-based model with non-deterministic sampling:**
>
> Addressing your concern on applicability, we integrated our steganographic module into another **DiT-based text-to-video model, LTX-Video** [4], and evaluated its performance. LTX-Video is a two-stage model using a more advanced EDM sampler than DDIM (when $\sigma_s \neq 0$, EDM sampler is non-deterministic). We inserted our steganography module into the diffusion process of the second stage. Due to differences between Wan2.1 and LTX-Video, we re-conducted hyperparameter experiments and determined the following hyperparameter values by testing on a small number of samples: $k_s=1.1,\text{cfg}=1.8,\tau_1=0.32,\tau_2=0.02$. The results, in the table below, demonstrate that our method also maintains strong performance on EDM-based video models.
>
> | Method | Accuracy (%)↑ | PSNR (dB)↑ | BRISQUE ↓ | Capacity (bits)↑ |
> | --- | --- | --- | --- | --- |
> | ours (Wan2.1) | 99.17 | 41.66 | 28.90 | 11983 |
> | ours (LTX-Video) | 99.23 | 41.39 | 34.21 | 4281 |
>
> Note that LTX-Video's video quality is slightly inferior to Wan2.1, which explains the higher BRISQUE scores of LTX-Video. Also, the latent size in LTX-Video is approximately 2/5 of Wan2.1’s latent size, leading to a reduction in capacity.
>
> `Weakness 2：Despite focusing on robustness, the method does not always guarantee full secret message reconstruction, even under clean conditions. This raises concerns about its practical reliability in high-stakes environments.`
>
> Your insightful question about the inability to fully reconstruct the secret message even under clean conditions highlights an inherent challenge in our method, which mainly stems from the following two factors:
>
> 1. **The VAE encoding and decoding process is highly nonlinear and irreversible**. The stego video is generated by decoding the latent variable $X_s$, and then re-encoded back to latent $X_s'$. This lossy process causes variations in the latent variables, leading to errors in message extraction. To address this, we designed the Invariance Mask $M_1$, aiming to identify regions of the latent space that remain stable through the VAE encoding-decoding process.
> 2. **The Invariance Mask $M_1$ only estimates the robust regions** for the latent variable with VAE encoding-decoding. In our method, to ensure the sender and receiver share the same mask, we perform a VAE encoding-decoding preprocessing on the clean latent $X_0$, using this as an approximation for the robust regions of $X_s$.
>
> Notably, our method already achieves around 99% message accuracy with a high capacity exceeding 10,000 bits. This level of accuracy and capacity supports the practical use of error-correcting codes to realize full secret message recovery.
> To this end, we adopt Low-Density Parity-Check (LDPC) coding for error correction, using the open-source Python library `ldpc`. By setting the code rate to 0.8, we successfully improve the message accuracy to 100% while maintaining a usable capacity of 9,000 bits.
>
> | Method | Accuracy (%)↑ | Capacity (bits)↑ |
> | --- | --- | --- |
> | ours | 99.17 | 11,983 |
> | ours+LDPC | 100 | 9,000 |
>
> `Question 1:To address concerns regarding generalizability and reliability, testing transferability to alternative generative backbones.`
>
> Thank you for your concerns regarding the generalizability of our method. Please refer to our response in weakness 1: we have integrated and tested our steganographic method on the DiT-based video generation model LTX-Video. The results demonstrate the good generalization capability of our method. Moreover, our method supports both deterministic and non-deterministic samplers, and is theoretically compatible with mainstream text-to-video generation models.
>
> `Question 2: Analyzing reconstruction failures to understand the inherent brittleness of the encoding scheme`
>
> The inability to achieve 100% accurate secret message reconstruction mainly stems from the irreversible and lossy nature of the VAE encoding-decoding process.
>
> To illustrate the distortion introduced by the VAE encoding-decoding process, we analyze the distribution of two variables. Specifically, $v_1$ is defined as the reconstruction error between **$X_0$** and its VAE-reconstructed latent **$X_0'$**, i.e., **$v_1 = L_1(X_0, X_0')$**. $v_2$ is defined as **$v_2=v_1(M_1)$**, representing the reconstruction error in regions selected by mask **$M_1$**. As shown, without any masking, approximately 31% of the positions in **$X_0$** experience a distortion greater than 0.1. Such a level of perturbation significantly interferes with accurate message extraction. After applying the mask **$M_1$**, it is observed that all positions with perturbations greater than 0.1 are effectively filtered out. $M_1$ significantly enhances the robustness of the steganographic method.
>
> | Variable | [0, 0.1) | [0.1, 0.2） | [0.2, 0.3) | [0.3, 1] |
> | --- | --- | --- | --- | --- |
> | $v_1$ | 1,454,410(69.4%) | 488,785(23.3%) | 121,738(5.8%) | 31,707(1.5%) |
> | $v_2$ | 670,925(100%) | 0 | 0 | 0 |
>
> Finally, we appreciate your interest in the generalizability and robustness of our method. If the paper is accepted, we will include additional experiments on new models in the final submission and add a thorough discussion on the causes of imperfect message reconstruction in the appendix.
>
> [1] Ldstega: Practical and robust generative image steganography based on latent diffusion models. *Proceedings of the 32nd ACM International Conference on Multimedia*. 2024.
>
> [2] PSyDUCK: Training-Free Steganography for Latent Diffusion. *arXiv preprint arXiv:2501.19172* (2025).
>
> [3] Pulsar: Secure steganography for diffusion models. *Proceedings of the 2024 on ACM SIGSAC Conference on Computer and Communications Security*. 2024.
>
> [4] Ltx-video: Realtime video latent diffusion. *arXiv preprint arXiv:2501.00103* (2024).

---

> ### Comment · Area_Chair_PuqW · 2025-08-08
>
> Dear Reviewer,
>
> Could you give some feedback on the rebuttal provided by the authors?
>
> Your AC

---

### Official Review · Reviewer_t9WQ · 2025-07-03

**Clarity:** 3
**Significance:** 3
**Originality:** 3
**Rating:** 5
**Confidence:** 4

**Summary:**

This paper proposes a video steganography framework named LD-RoViS. It targets latent diffusion models that employ deterministic sampling strategies by modulating the CFG scale at the final step of the diffusion process, thereby constructing a steganographic channel for information embedding. The authors introduce a multi-masking mechanism to identify locations in the latent space that are both robust to distortion and highly discriminative for the embedded information. Experimental results demonstrate that the method achieves high embedding capacity and extraction accuracy.

**Questions:**

A core claim of the paper is that its method is applicable to general deterministic diffusion models, but all experiments are based on a single model employing a Flow Matching sampling strategy. We would like to know if this method remains effective on other types of deterministic samplers (e.g., deterministic DDIM)? Given the differences in update dynamics across various samplers, we hope the authors can provide additional arguments or experimental evidence to support this generalization claim. For example, a brief theoretical analysis or preliminary experiments on a publicly available DDIM-based model would greatly enhance our confidence in the method's scope of applicability.

To resist compression distortion, the multi-masking mechanism tends to embed information in texture-rich, high-frequency regions. However, these regions are typically also areas closely monitored by steganalysis tools. We would like to discuss with the authors whether this strategy of selecting embedding locations for the sake of robustness might inadvertently concentrate modification traces in specific areas, thereby forming new statistical features that could potentially be exploited by future, more advanced analysis models?

**Ethical Concerns:**

["NO or VERY MINOR ethics concerns only"]

**Final Justification:**

The author's feedback has addressed some of my concerns, including doubling the dataset size and adding another model for evaluation. I have decided to retain the original score.

**Limitations:**

Yes.

**Quality:**

3

**Strengths And Weaknesses:**

Strengths:

The paper extends the applicability of generative steganography to deterministic diffusion models. Currently, many generative steganography techniques rely on interfering with stochastic sampling processes, which prevents their direct application to increasingly popular deterministic samplers. This paper addresses this by utilizing the CFG scale, a universal parameter, to construct the steganographic channel.

The multi-masking mechanism proposed in the paper is a noteworthy component. This mechanism constructs an "invariance mask" by explicitly simulating the encoding-decoding and noise injection processes, and combines it with a "discriminability mask" generated by CFG modulation. Its effectiveness is supported by data in the Ablation Studies, demonstrating the necessity of both mask components for achieving high extraction accuracy.

Weaknesses:

The paper lacks sufficient ablation studies regarding the design choices for the core CFG modulation strategy (e.g., why modulation occurs in the final step, why symmetric perturbation is used).

Performance evaluation is based on **only 100 video samples.** For a method aiming to handle diverse video content, this scale needs further improvement.

The experimental evidence presented in the paper originates from **a single model only**.

Similar to all generative methods, the upper bound of the paper's method performance (including visual quality and imperceptibility) is closely coupled with the performance of the chosen base video generation model; any flaws in the base model will directly transfer to the steganographic video.

---

> ### Author Rebuttal · Authors · 2025-07-30
>
> We are pleased to receive your positive feedback. Thank you for recognizing our contribution in extending steganography to deterministic diffusion models. Below, we provide point-by-point responses to the weaknesses you listed.
>
> `Weakness 1：The paper lacks sufficient ablation studies regarding the design choices for the core CFG modulation strategy (e.g., why modulation occurs in the final step, why symmetric perturbation is used).`
>
> We provide additional analysis below to clarify our design choices for the CFG modulation strategy.
>
> - **Why modulation is applied at the final denoising step:**
>
> Following your suggestion, we conducted ablation studies by applying the modulation at different denoising steps. The results are presented in the table below. Modulating at **earlier steps negatively affects message extraction accuracy**, as evidenced by both lower accuracy and higher standard deviation with increasing $t$. We hypothesize that when parameters are modulated at early denoising steps, **the resulting latent variables $X_0^1$ and $X_0^2$ become more uncontrollable.** After several denoising steps, the difference between $X_0^1$ and $X_0^2$ may either increase or decrease unpredictably. As our message extraction relies heavily on the distinguishability between $X_0^1$ and $X_0^2$, this uncertainty leads to lower extraction accuracy. Meanwhile, PSNR, **BRISQUE scores, and steganalysis results remain stable**, indicating that the perceptual quality of the video does not degrade significantly. Moreover, the stego videos are statistically indistinguishable from clean videos, suggesting a high level of imperceptibility. Based on the above results, we keep the strategy of applying parameter modulation at the final denoising step to ensure accurate message extraction.
>
> | Time step $t$ | Accuracy (%)↑ | PSNR (dB)↑ | BRISQUE↓ | CovNet $P_e$(%)↑ |
> | --- | --- | --- | --- | --- |
> | $t$=1 (Ours) | 99.17+-0.63 | 41.66 | 28.90 | 49.74 |
> | $t$=2 | 98.43+-1.12 | 41.22 | 29.99 | 49.93 |
> | $t$=4 | 95.72+-2.57 | 41.34 | 28.06 | 49.78 |
> | $t$=8 | 94.82+-3.30 | 41.94 | 29.62 | 49.72 |
>
> - **Why symmetric perturbation is used:**
>
> This is a practical simplification. Since message extraction relies on the difference between $X_0^1$ and $X_0^2$, which is determined by the distance between $k_1$ and $k_2$, we only care about their relative distance. Therefore, we define $k_1 = k_s - \text{cfg}, k_2 = k_s + \text{cfg}$, using symmetric perturbation to **reduce the number of tunable parameters** and make the method more deployment-friendly in practice.
>
> Considering that $k_1$ and $k_2$ in our method are essentially controlled by a single hyperparameter $\text{cfg}$, we will remove the use of $k_1$ and $k_2$ in the final submission and retain only the hyperparameter $\text{cfg}$. Meanwhile, to avoid confusion between the hyperparameter $\text{cfg}$ and the commonly used CFG scale in diffusion models, we will rename $\text{cfg}$ to $k_m$ to reduce ambiguity.
>
> `Weakness 2：Performance evaluation is based on only 100 video samples. For a method aiming to handle diverse video content, this scale needs further improvement.`
>
> Following previous works on video steganography, such as [1], we initially evaluated our method on 100 video pairs. Although this dataset yields **8100 frame pairs**, which is sufficient for standard image-level steganalysis, we acknowledge that the size of the video dataset could be further improved to better reflect real-world diversity.
>
> In response to your suggestion, we have generated and evaluated an additional 100 video pairs, resulting in a total of **200 video pairs**. The experimental results, summarized below, demonstrate that the performance of our method remains stable with the larger dataset, indicating good generalization. These results show minimal variation in metrics, validating that our method performs consistently even with a doubled dataset.
>
> | Dataset Size | Accuracy (%)↑ | PSNR (dB)↑ | BRISQUE ↓ | Capacity (bits)↑ | CovNet $P_e$ (%) ↑ |
> | --- | --- | --- | --- | --- | --- |
> | 100 Videos (original) | 99.17 ± 0.63 | 41.66 ± 1.52 | 28.90 ± 6.05 | 11983 ± 1446 | 49.74 |
> | 200 Videos (extended) | 99.19 ± 0.56 | 42.08 ± 1.88 | 28.99 ± 4.50 | 11704 ± 1085 | 49.78 |
>
> `Weakness 3：The experimental evidence presented in the paper originates from a single model only.`
>
> Due to the significantly higher computational cost and time required for video generation and testing compared to images, our initial submission evaluated performance solely on one current mainstream video generation model, Wan2.1.
>
> To address your concern about generalization, we conducted preliminary experiments on another video generation model, LTX-Video [2]. LTX-Video is a two-stage model utilizing a more advanced EDM sampler compared to DDIM. We inserted our steganography module into the diffusion process of the second stage. Due to differences between Wan2.1 and LTX-Video, we re-conducted hyperparameter experiments and determined the following hyperparameter values by testing on a small number of samples: $k_s=1.1,\text{cfg}=1.8,\tau_1=0.32,\tau_2=0.02$.
>
> | Method | Accuracy (%)↑ | PSNR (dB)↑ | BRISQUE ↓ | Capacity (bits)↑ |
> | --- | --- | --- | --- | --- |
> | ours (Wan2.1) | 99.17 | 41.66 | 28.90 | 11983 |
> | ours (LTX-Video) | 99.23 | 41.39 | 34.21 | 4281 |
>
> The results show that our method maintains strong performance on a video model based on the EDM sampler. Note that LTX-Video's video quality is slightly inferior to Wan2.1, which explains the higher BRISQUE scores of LTX-Video. Also, the latent size in LTX-Video is approximately 2/5 of Wan2.1’s latent size, leading to a reduction in capacity.
>
> `Weakness 4：Similar to all generative methods, the upper bound of the paper's method performance (including visual quality and imperceptibility) is closely coupled with the performance of the chosen base video generation model; any flaws in the base model will directly transfer to the steganographic video.`
>
> We acknowledge that the performance of generative steganography methods depends to some extent on the quality of the chosen generative model. However, this does not compromise the practical utility of our steganographic method. The essence of steganography lies in **behavioral security — being indistinguishable from normal behavior.** And the widespread use of generated videos on social media platforms has made them a common and accepted form of content. As long as the stego videos are indistinguishable from normal generated videos, behavioral-level security can still be achieved. Although stego videos may present certain flaws, similar limitations are often observed in normally generated videos as well, rendering them hard to differentiate from this viewpoint.
>
> Moreover, our method exhibits strong generalizability across different generative models. As the performance of generative models continues to improve, the visual quality of the stego video of our method is also expected to benefit accordingly.
>
> `Question 1：We would like to know if this method remains effective on other types of deterministic samplers (e.g., deterministic DDIM)? `
>
> As explained in our response to Weakness 3, we have validated the effectiveness of our method on the EDM sampler-based LTX-Video model. Meanwhile, our method supports both deterministic and non-deterministic sampling processes. **This is due to our parameter modulation strategy, which only needs to modify the CFG scale parameter during the final denoising step.** To the best of our knowledge, mainstream text-to-video generation models all support the CFG scale parameter, indicating that our method is broadly applicable and is theoretically fully compatible with samplers such as deterministic DDIM.
>
> `Question 2：We would like to discuss with the authors whether this strategy of selecting embedding locations for the sake of robustness might inadvertently concentrate modification traces in specific areas, thereby forming new statistical features that could potentially be exploited by future, more advanced analysis models?`
>
> According to the study in [3], in the denoising process of diffusion models, **earlier steps typically reconstruct low-frequency structures and semantics**, while **later steps capture high-frequency details and textures**. Our steganographic modulation is applied at the final denoising step, and we employ a multi-mask strategy to select robust regions in the latent space. Together, these designs result in stego modifications being concentrated in high-frequency areas of the image, which typically correspond to fine-grained textures, as illustrated in Figure 3 of our main paper.
>
> According to the theory of adaptive steganography [4], **high-frequency regions in video** are more information-dense and closer to randomness, making it inherently more difficult for stego modifications in these areas to be accurately detected. This provides theoretical support for the security of our method.
>
> Furthermore, our steganalysis experiments show that neither the video codec-based SUPERB method nor deep learning-based methods such as CovNet and LWENet are able to effectively detect our steganographic traces. This suggests that the modifications introduced by the multi-mask strategy do not produce statistically significant anomalies, providing strong empirical evidence for the security of our method.
>
> [1] Adaptive QIM with minimum embedding cost for robust video steganography on social networks. IEEE Transactions on Information Forensics and Security 17 (2022): 3801-3815.
>
> [2] Ltx-video: Realtime video latent diffusion. *arXiv preprint arXiv:2501.00103* (2024).
>
> [3] Boosting diffusion models with moving average sampling in frequency domain. *Proceedings of the IEEE/CVF conference on computer vision and pattern recognition*. 2024.
>
> [4] Content-adaptive steganography by minimizing statistical detectability. *IEEE transactions on information forensics and security* 11.2 (2015): 221-234.

---

> ### Comment · Area_Chair_PuqW · 2025-08-08
>
> Dear Reviewer,
>
> Could you give some feedback on the rebuttal provided by the authors?
>
> Your AC

---

### Official Review · Reviewer_tdSw · 2025-07-04

**Clarity:** 2
**Significance:** 2
**Originality:** 3
**Rating:** 4
**Confidence:** 3

**Summary:**

This work proposes a generative video steganography method for latent diffusion models by proposing an implicit parameter modulation strategy with keys and a multi-mask mechanism for identifying robust regions. It was demonstrated that this work can embed 12,000 bits of secret messages into a 5-second video with an accuracy >99%.

**Questions:**

Please address the major concerns above.

It is unclear if multi-mask is robust for any video contents. Any discussion?

**Ethical Concerns:**

["NO or VERY MINOR ethics concerns only"]

**Final Justification:**

The rebuttal has addressed most of my concerns. After reading the rebuttal and other reviews, I will increase my score.

**Limitations:**

Yes

**Quality:**

3

**Strengths And Weaknesses:**

Strengths:
1) This work addresses an interesting problem of generative video steganography for modern video generation models based on DDIM.
2) Evaluating for video compression seems practical.

Weaknesses:
1) Even though this work claimed that it is for video, theoretically the method for LDM for image generation can be applicable to LDM for video generation by replacing the decoder. Thus, in terms of novelty, it is unclear if this work has specific novelty over other generative image steganography methods with LDM. See [33], [36] as well as the followings:
- Improved Generative Steganography Based on Diffusion Model (10.1109/TCSVT.2025.3539832)
- Conditional Diffusion Model for Image Steganography (10.1109/AIPMV62663.2024.10692262)
- Establishing Robust Generative Image Steganography via Popular Stable Diffusion (10.1109/TIFS.2024.3444311)
- Generative Image Steganography Based on Text-to-Image Multimodal Generative Model (10.1109/TCSVT.2025.3556892)
- DGADM-GIS: Deterministic Guided Additive Diffusion Model for Generative Image Steganography (10.1109/TDSC.2025.3578676)
- Diffusion-Stego: Training-free diffusion generative steganography via message projection (10.1016/j.ins.2025.122358)
- Provably Secure Covert Messaging Using Image-Based Diffusion Processes (10.1109/SaTML64287.2025.00057)
- StegaFDS: Generative Steganography Based on First-Order DPM-Solver (10.1109/TrustCom63139.2024.00042)
- Conditional Flow-based Generative Steganography (10.1109/TDSC.2025.3570468)
Since the proposed method looks generic for any LDM, it seems important to comprehensively discuss, compare and demonstrate over these prior works. Moreover, the proposed parameter modulation strategy looks very similar to the above works, so it is important to distinguish the proposed method over them in the setting of LDM.
2) Even though 12,000 bits of data into a 5-sec video may be great, it seems also possible to embed messages per image using any above image-based methods, which can embed 1 bpp with 100% accuracy - of course it may fail for video compression. However, there is no demonstration for it, so it is difficult to see the proposed method as a SOTA. Thus, the proposed method should be compared with other image-based methods as well as other video steganography method if possible, such as
- Large-capacity and Flexible Video Steganography via Invertible Neural Network (CVPR 2023)
- StegaNeRV: Video Steganography using Implicit Neural Representation (CVPRW 2024)
- Efficient three layer secured adaptive video steganography method using chaotic dynamic systems (Scientific Reports 2024)

---

> ### Author Rebuttal · Authors · 2025-07-30
>
> Thank you for the time you have devoted and for your valuable suggestions. Below, we will address each of the weaknesses you listed individually.
>
> `Weakness 1：In terms of novelty, it is unclear if this work has specific novelty over other generative image steganography methods with LDM. Since the proposed method looks generic for any LDM, it seems important to comprehensively discuss, compare and demonstrate over these prior works. Moreover, the proposed parameter modulation strategy looks very similar to the above works, so it is important to distinguish the proposed method over them in the setting of LDM.`
>
> We sincerely thank the reviewer for providing detailed references and raising this insightful question. We systematically compared our method with existing works based on three key dimensions: **whether inversion is required**, **whether the method supports latent diffusion models**, and **whether it is compatible with deterministic samplers**. The results are shown in the table.
>
> | Method | Inversion-Free | Latent Diffusion Compatible | Deterministic Sampler Compatible |
> | --- | --- | --- | --- |
> | [1], [2], [4], [5], [6], [8], [9] | × | × | √ |
> | [3], [7] | × | √ | √ |
> | [10], [11], [12] | √ | √ | × |
> | ours | √ | √ | √ |
> - **Novelty compared to LDM-based image steganography**
>
> We observe that works [1]–[9] hide messages in the initial latent $X_T$﻿ of the denoising process and rely on inversion at the receiver side to reconstruct $X_T'$﻿ from the stego image. However, most diffusion samplers (e.g., DDIM, Flow Matching) are designed for one-directional generation and **cannot achieve perfect invertibility**. For example, method [1] retrains DDIM to improve invertibility, method [7] introduces a reversible sampler (EDICT), and method [9] employs a reversible flow-based model instead of a diffusion model.
>
> On the other hand, methods [10]–[12] embed messages in the random noise of the final denoising step. However, these methods rely on non-deterministic sampling (e.g., DDPM or DDIM with $\sigma_s\ne 0$), which is not compatible with most modern video generation models that employ deterministic samplers (e.g., Flow Matching or DDIM with $\sigma_s=0$﻿).
>
> Our work is explicitly designed to overcome these limitations by proposing a steganographic method that is **inversion-free**, **compatible with latent diffusion models**, and **suitable for deterministic samplers**.
>
> - **How our parameter modulation strategy differs from previous works**
>
> We leverage the CFG scale, an attribute control factor in text-to-video models, to embed secret information in the final denoising step, eliminating the reliance on non-deterministic samplers. This enables high-capacity steganography while maintaining high video quality. To the best of our knowledge, **this is the first generative steganography method based on CFG scale**, offering both generality and practicality under latent diffusion models.
>
> `Weakness 2：The proposed method should be compared with other image-based methods as well as other video steganography method if possible.`
>
> - **Comparison with the provided video steganography methods**
>
> We have carefully studied the three mentioned video steganography works. Works [13] and [14] are designed to hide **images or video frames** within videos. Work [15] is a spatial-domain adaptive steganography method. In fact, works [13]–[15] fall into the category of modification-based video steganography, where message embedding requires alterations to video content, making them vulnerable to both distortion drift and steganalysis attacks.
>
> - **Comparison with the provided image steganography methods**
>
> Regarding comparisons with image steganography methods: most mentioned diffusion-based image steganography methods ([1]–[12]) either rely heavily on **exact inversion** or require **non-deterministic samplers**, making them incompatible with state-of-the-art video generation models like Wan2.1. Nonetheless, in response to your suggestion, we implemented a variant of [10] (LDStega) adapted to video diffusion models, referred to as **LDStega_video**. To simulate the stochasticity required by [10], **we manually injected Gaussian noise into the final denoising step**—an operation that does **not exist by default in deterministic samplers**. To simulate a realistic non-deterministic sampling process, we adopt the noise standard deviation from the final denoising step of the **non-deterministic sampler EDM Sampler [16]**, which is set to **$\sigma = 2\times10^{-3}$**. For fair comparison, we also created a low-capacity version (**LDStega_adjust**) with 10,080 bits embedded per video.
>
> The results are shown in the table below. We observe that:
>
> **(1) The accuracy** of LDStega variants is significantly lower than ours;
>
> **(2) Visual quality** (PSNR and BRISQUE) degrades heavily for LDStega_video, and it is easily detected by the CovNet steganalyzer;
>
> (3) Even the tuned LDStega_adjust performs worse than our method across all metrics.
>
> | Method | Accuracy (%) ↑ | PSNR ↑ | BRISQUE ↓ | Capacity (bits) | CovNet Pₑ (%) ↑ |
> | --- | --- | --- | --- | --- | --- |
> | Ours | 99.17 | 41.66 | 28.90 | 11,983 | 49.74 |
> | LDStega_video | 84.50 | 29.09 | 43.42 | 2,096,640 | 0.01 |
> | LDStega_adjust | 84.97 | 36.51 | 34.06 | 10,080 | 37.13 |
>
> This performance gap can be attributed to two key factors:
>
> **(1) Video latent diffusion models are more compressed** than image models. For example, Wan2.1’s latent space has a 64× compression ratio compared to the original video, making message recovery more difficult.
>
> **(2) Temporal correlations in video make steganographic distortions more noticeable**. While a small perturbation might be tolerable in a static image, it could cause significant temporal artifacts in video, leading to perceptual degradation or easier detection.
>
> Therefore, even when compared with state-of-the-art image steganography methods, our proposed video steganography method demonstrates clear advantages in terms of **robustness, quality preservation, and steganalysis resistance**.
>
> Finally, we wish to clarify that our method is **not tied to a specific sampler**. Since the embedding relies only on modulating the **CFG scale** in the final denoising step—a parameter widely supported in modern diffusion models—our method is broadly compatible with **DDPM, DDIM, Flow Matching**, and other popular sampling frameworks with CFG scale.
>
> Below are the answers to the questions you raised:
>
> `Question: It is unclear if multi-mask is robust for any video contents. Any discussion?`
>
> We believe that our proposed **multi-mask** strategy is robust for diverse video content, as supported by both experimental results and its underlying design:
>
> First, the **experimental results provide strong empirical evidence**. Our steganographic framework achieves **over 99% message extraction accuracy** on a diverse test set of **100 videos with different prompts**, which demonstrates its applicability across a wide range of video contents.
>
> Second, from a design perspective, the **multi-mask strategy is based on latent-space percentile selection**, rather than relying on pixel-level or semantic information. The Invariance Mask $M_1$ identifies the $\tau_1$% **least-different positions** between $X_0$ and $X_0'$, while the Discriminative Mask $M_2$ captures the $\tau_2$% **most-different positions** between $X_0^1$ and $X_0^2$. $M=M_1\odot M_2$ indicates positions meeting both conditions. Regardless of the variations in video content, our multi-mask strategy always selects the most robust regions in the latent space for message embedding, ensuring the robustness of the proposed method.
>
> Finally, we truly appreciate your pointers to numerous related works. If the paper is accepted, we will include a more comprehensive discussion and comparison with these works in the final submission to better highlight the novelty of our method.
>
> [1] Improved Generative Steganography Based on Diffusion Model (10.1109/TCSVT.2025.3539832)
>
> [2] Conditional Diffusion Model for Image Steganography (10.1109/AIPMV62663.2024.10692262)
>
> [3] Establishing Robust Generative Image Steganography via Popular Stable Diffusion (10.1109/TIFS.2024.3444311)
>
> [4] Generative Image Steganography Based on Text-to-Image Multimodal Generative Model (10.1109/TCSVT.2025.3556892)
>
> [5] DGADM-GIS: Deterministic Guided Additive Diffusion Model for Generative Image Steganography (10.1109/TDSC.2025.3578676)
>
> [6] Diffusion-Stego: Training-free diffusion generative steganography via message projection (10.1016/j.ins.2025.122358)
>
> [7] Provably Secure Covert Messaging Using Image-Based Diffusion Processes (10.1109/SaTML64287.2025.00057)
>
> [8] StegaFDS: Generative Steganography Based on First-Order DPM-Solver (10.1109/TrustCom63139.2024.00042)
>
> [9] Conditional Flow-based Generative Steganography (10.1109/TDSC.2025.3570468)
>
> [10] Ldstega: Practical and robust generative image steganography based on latent diffusion models. *Proceedings of the 32nd ACM International Conference on Multimedia*. 2024.
>
> [11] PSyDUCK: Training-Free Steganography for Latent Diffusion. *arXiv preprint arXiv:2501.19172* (2025).
>
> [12] Pulsar: Secure steganography for diffusion models. *Proceedings of the 2024 on ACM SIGSAC Conference on Computer and Communications Security*. 2024.
>
> [13] Large-capacity and Flexible Video Steganography via Invertible Neural Network (CVPR 2023)
>
> [14] StegaNeRV: Video Steganography using Implicit Neural Representation (CVPRW 2024)
>
> [15] Efficient three layer secured adaptive video steganography method using chaotic dynamic systems (Scientific Reports 2024)
>
> [16] Ltx-video: Realtime video latent diffusion. *arXiv preprint arXiv:2501.00103* (2024).

---

> ### Comment · Area_Chair_PuqW · 2025-08-08
> **Feedback**
>
> Dear Reviewer,
>
> Could you give some feedback on the rebuttal provided by the authors?
>
> Your AC

---

### Note · Authors · 2025-08-12

We thank the reviewers and the AC for their thoughtful comments. During the rebuttal phase, we addressed all concerns raised in the initial reviews, and no new questions were posed by any reviewer afterwards. We believe this reflects that our clarifications and additional results have satisfactorily resolved all outstanding issues.

In our work, we propose a **training-free and robust video steganography framework for deterministic latent diffusion models**. By modulating implicit conditional parameters during the diffusion process and introducing a novel **multi-mask mechanism**, we construct a steganographic channel that is highly robust to video compression. Our method can embed ~12,000 bits in a 5-second video with extraction accuracy exceeding 99%.

In response to reviewer feedback:

1. **Generality across models:** We extended our experiments to another video generative model **LTX-Video**, confirming that our method maintains its high performance.
2. **Novelty validation:** We conducted theoretical and experimental comparisons with leading image-based generative steganography methods, demonstrating state-of-the-art performance.
3. **Effectiveness of multi-mask mechanism:** We provided detailed theoretical analysis and quantitative evidence supporting its role in enhancing robustness.
4. **Ablation completeness:** Additional experiments confirmed that our chosen configuration is optimal for robustness and capacity.

Reviewers consistently recognized that our method offers **strong resilience against video compression**, **practical applicability**, and **extends the applicability of generative steganography to deterministic diffusion models**. We respectfully encourage the AC to consider these strengths, the positive reception from reviewers, and the absence of unresolved concerns when making the final decision.

We sincerely thank the committee for their time and consideration.

---

### Decision · Program_Chairs · 2025-09-17

**Decision:**

Accept (poster)

**Comment:**

After the rebuttal, the paper received ratings of 4455. The paper presents a training-free video steganography framework to enhance robustness against compression artifacts in deterministic latent diffusion models. It introduces a new multi-mask mechanism for embedding information, which achieves an impressive capacity of about 12,000 bits with over 99% extraction accuracy. The presented approach is practical with strong performance. There were some concerns regarding limited initial evaluations and generalizability, but the authors have addressed reviewers' concerns well. Given the positive feedback from the reviewers, the AC recommends acceptance to NeurIPS 2025.